# Crop Height Estimation of Corn from Multi-Year RADARSAT-2 Polarimetric Observables Using Machine Learning

Qinghua Xie [1,2,3], Jinfei Wang [3,4], Juan M. Lopez-Sanchez [2], Xing Peng [1,*], Chunhua Liao [3], Jiali Shang [5], Jianjun Zhu [6], Haiqiang Fu [6] and J. David Ballester-Berman [2]

1 School of Geography and Information Engineering, China University of Geosciences (Wuhan), Wuhan 430074, China; xieqh@cug.edu.cn
2 Institute for Computer Research (IUII), University of Alicante, E-03080 Alicante, Spain; juanma.lopez@ua.es (J.M.L.-S.); davidb@ua.es (J.D.B.-B.)
3 Department of Geography and Environment, The University of Western Ontario, London, ON N6A 5C2, Canada; jfwang@uwo.ca (J.W.); cliao33@uwo.ca (C.L.)
4 Institute for Earth and Space Exploration, The University of Western Ontario, London, ON N6A 3K7, Canada
5 Ottawa Research and Development Centre, Agriculture and Agri-Food Canada, Ottawa, ON K1A 0C6, Canada; jiali.shang@canada.ca
6 School of Geosciences and Info-Physics, Central South University, Changsha 410083, China; zjj@csu.edu.cn (J.Z.); haiqiangfu@csu.edu.cn (H.F.)
* Correspondence: pengxing@cug.edu.cn

**Abstract:** This study presents a demonstration of the applicability of machine learning techniques for the retrieval of crop height in corn fields using space-borne PolSAR (Polarimetric Synthetic Aperture Radar) data. Multi-year RADARSAT-2 C-band data acquired over agricultural areas in Canada, covering the whole corn growing period, are exploited. Two popular machine learning regression methods, i.e., Random Forest Regression (RFR) and Support Vector Regression (SVR) are adopted and evaluated. A set of 27 representative polarimetric parameters are extracted from the PolSAR data and used as input features in the regression models for height estimation. Furthermore, based on the unique capability of the RFR method to determine variable importance contributing to the regression, a smaller number of polarimetric features (6 out of 27 in our study) are selected in the final regression models. Results of our study demonstrate that PolSAR observables can produce corn height estimates with root mean square error (RMSE) around 40–50 cm throughout the growth cycle. The RFR approach shows better overall accuracy in corn height estimation than the SVR method in all tests. The six selected polarimetric features by variable importance ranking can generate better results. This study provides a new perspective on the use of PolSAR data in retrieving agricultural crop height from space.

**Keywords:** crop height; RADARSAT-2; corn; Synthetic Aperture Radar (SAR); PolSAR; machine learning; RFR; SVR; agriculture

## 1. Introduction

Crop height is an important agronomic descriptor related to crop type, biomass estimation, phenological stage, potential yield, detection of growth anomalies (e.g., diseases, pests, weather disasters, and cereal lodging), and precision fertilization [1–3]. Traditional methods to monitor crop height by visual inspection require a huge workforce over large areas [4]. Synthetic Aperture Radar (SAR), with its capability of imaging in day and night and all weather conditions and its sensitivity to the geometric and physical properties of the target, has shown to be an effective remote sensing technique in crop biophysical parameter retrieval at regional and global scales. For crop height estimation with SAR observations, the approaches can be generally categorized into three types: backscattering model methods, interferometric SAR (InSAR) methods, and data-driven empirical model methods [1].

The backscattering models for crop height retrieval include both physical models and semi-empirical models. The physical model developed for crop height usually adopts a discrete approach, such as the Radiative Transfer Theory Model (RTM), which is able to simulate the backscattering coefficient for crop targets from the perspective of fine electromagnetic scattering as a function of various geometric and physical parameters of the plant, such as canopy height, dielectric constant, number of layers and leaves, leaf angle and size, stem width, and so on [5–12]. The physical scattering model usually depends on the polarization and crop type. Due to the complexity of the physical scattering model, crop height estimation may require computationally expensive Monte Carlo simulations to relate the SAR measurements to parameters describing the entire canopy's physical characteristics [1]. Moreover, the inversion process of model parameters often leads to ill-posed problems due to a high-dimensional parameter space [1,4,13,14]. Although the merging of a metamodel (e.g., the polynomial chaos expansion (PCE)) with the backscattering model enables a significant reduction of the computational cost and the complexity involved in the inversion scheme, the growth stage needs to be identified in advance to narrow the solution space [4,13,14]. For vegetation, the most popular semi-empirical model is the Water Cloud Model (WCM) proposed by Attema and Ulaby [15]. Due to its simplicity and practicability, the WCM has been extensively applied to soil moisture estimation and to the retrieval of various vegetation biophysical variables, such as leaf area index (LAI), aboveground biomass, and vegetation height [16–18]. However, the retrieval results from the original WCM often show low accuracies attributed to many assumptions and simplifications involved in the model. In past studies, many modifications of the WCM have been developed by considering more complex scattering mechanisms or more vegetation geometrical properties. Moreover, there were some studies reporting calibration of the model coefficients of the WCM for specific areas, vegetation types, or SAR sensors based on some ground measurements [19].

The InSAR method exploits the interferometric phase between two co-registered SAR images acquired in the same polarization to capture the height of the scattering phase center [20], which is approximately considered as the crop surface height in agricultural areas, like a digital surface model. Then, an external digital terrain model (DTM), also called vegetation-free digital elevation model (DEM), is required to derive the crop height itself [1]. In order to obtain accurate crop surface height, it needs some strict conditions in general. For example, the available SAR data is expected to work with short wavelengths and appropriate polarization to enable the scattering phase center to be located as close as possible to the top of the canopy. For the same purpose, a structurally dense crop is required as well. Since the crop height is usually very low with respect to forest height, a relatively large spatial baseline is required to reduce the height of ambiguity. Moreover, a short enough revisit time is expected to obtain high coherence since crop height is assumed unchanged within this time interval. In addition, the quality of the DTM data used to remove the underlying topography from crop surface height is also an unignorable factor. Although recently, single-polarization InSAR data at the L- and P-band have proven its capability to generate a comparable performance in DTM inversion in forest areas with respect to the traditional PolInSAR method with fully polarimetric SAR data [21], currently, the accurate DTM product over a vegetation area is mostly generated from other measurement technologies, such as light detection and ranging(LIDAR), polarimetric SAR interferometry (PolInSAR) [22–24], SAR Tomography (TomoSAR) [25], and field topographic mapping. The PolInSAR method combines the interferometric and polarimetric information to better separate the different scattering phase centers in the vegetation volume, which has been demonstrated to be useful for estimating vegetation structural parameters [26,27]. For vegetation height estimation, the PolInSAR technique has been validated in a variety of forest types with many airborne and few spaceborne datasets at different radar frequencies [22,23,28–39]. The performance is constrained by two key aspects: temporal decorrelation and spatial baseline. A shorter revisit time is expected to provide higher coherence, which is related to the quality of the interferometric phase [40]. A relatively

large baseline is expected to provide enough sensitivity of height measurement. Since crops grow faster and have shorter heights than forests, the availability of PolInSAR data for crop height estimation is more constrained than for forest height inversion [1–3]. Until now, few successful examples of crop height retrieval with the PolInSAR method are restricted to data acquired in indoor experiments [41,42] and airborne campaigns [43,44]. With regard to spaceborne datasets, several authors have reported successful results with a dataset over paddy rice fields acquired from the science phase of the TanDEM-X mission (bistatic configuration) from April to September 2015, being the baselines especially adjusted to ten times the regular configuration, i.e., around 2–3 km [1,3]. Although the PolInSAR method shows the capability to produce accurate estimates of crop height, currently the available PolInSAR datasets for crop height monitoring are very limited.

Another available option to estimate crop height is the data-driven empirical model method, which in this study refers to the use of a regression approach to train an empirical model between some PolSAR observables and crop height. The unknown crop heights in a scene are predicted by the trained regression model and the corresponding PolSAR observables. A large number of previous studies have been reported to investigate the correlations between PolSAR observables (e.g., backscattering coefficients, polarimetric decomposition parameters, radar vegetation index, and correlation coefficients), and crop parameters (e.g., LAI, PAI, biomass, phenological stages, canopy coverage, and crop height) over different crop fields at different radar frequencies [45–54]. These research results have shown the potential of PolSAR observables for crop parameter retrieval. However, to date, there are few studies reporting crop height retrieval based on regression with PolSAR observables [55–57]. In these studies, due to the limitation of images available and field data collected, the volume of available observed samples for training is in general relatively small. The crop types studied are also limited, such as sunflower, wheat, and canola. Moreover, the number and types of selected PolSAR observables are limited, and an empirical relationship model or linear/polynomial regression model is usually chosen [55–57].

Due to the aforementioned limitations of the backscattering model methods and interferometric SAR methods in practice, this study is focused on data-driven empirical model methods. More specifically, the present study aims at providing a comprehensive demonstration and validation of crop height retrieval of corn by exploiting a large number of PolSAR observables and ground measurements with machine learning regression methods. A large dataset formed by multi-temporal C-band RADARSAT-2 (launched by the Canadian Space Agency) images and quasi-synchronous in situ measurements of crop height along three years over two geographically close study sites in Canada, with similar agricultural practices and climatic conditions, are exploited here. Corn has been chosen as the target crop for this experiment because it has significant socio-economic interest for humans' staple food, a raw material of ethanol and animal feed, and it is the cereal with the highest production worldwide. Moreover, the height range of corn during the whole growing period is relatively large, reaching over 3 m in our study at the final stage, which results in more radar signature differences within the time-series SAR acquisitions. In addition, radar response to corn with RADARSAT-2 data have been well studied in previous literature [42,45,46,48,50,58–60]. Two typical machine learning regression methods, i.e., Random Forest Regression (RFR) and Support Vector Regression (SVR), are adopted for model training instead of linear regression. Moreover, the unique ability of the RFR method to provide variable importance contributing to the regression can help us understand the results and further investigate the performance after filtering the selection of input PolSAR observables.

## 2. Materials and Methods

### 2.1. Study Site and Dataset

As shown in Figure 1, two geographically close sites both located in Southwestern Ontario, Canada, were selected. One site was located in the west of London, and the other

one was near Stratford. Both study sites were agricultural areas including mainly crop fields, a few buildings, and forests. Corn, wheat, and soybean were the dominant crop types in both study sites. There was also some alfalfa, hay, and grass growing in these areas. Both study sites were suitable for cultivating crops because of abundant precipitation, mild weather, and productive soil, with relatively flat topography. In both study areas, corn and soybean were seeded in May and harvested in October of the same year. In contrast, the winter wheat growing period crossed over two years, seeded in October and harvested in July of the following year. It should be noted that in both study areas, crop rotation did not need to be done in the same field. The farmers' practice was to retain residuals for soil conservation. Therefore, one crop field may harvest residuals from another crop field from a previous year. For example, the cornfields may have residuals of wheat or soybean. All cornfields employed for height inversion and the locations of the sample points for collecting ground measurements are marked in green and red in Figure 1, respectively.

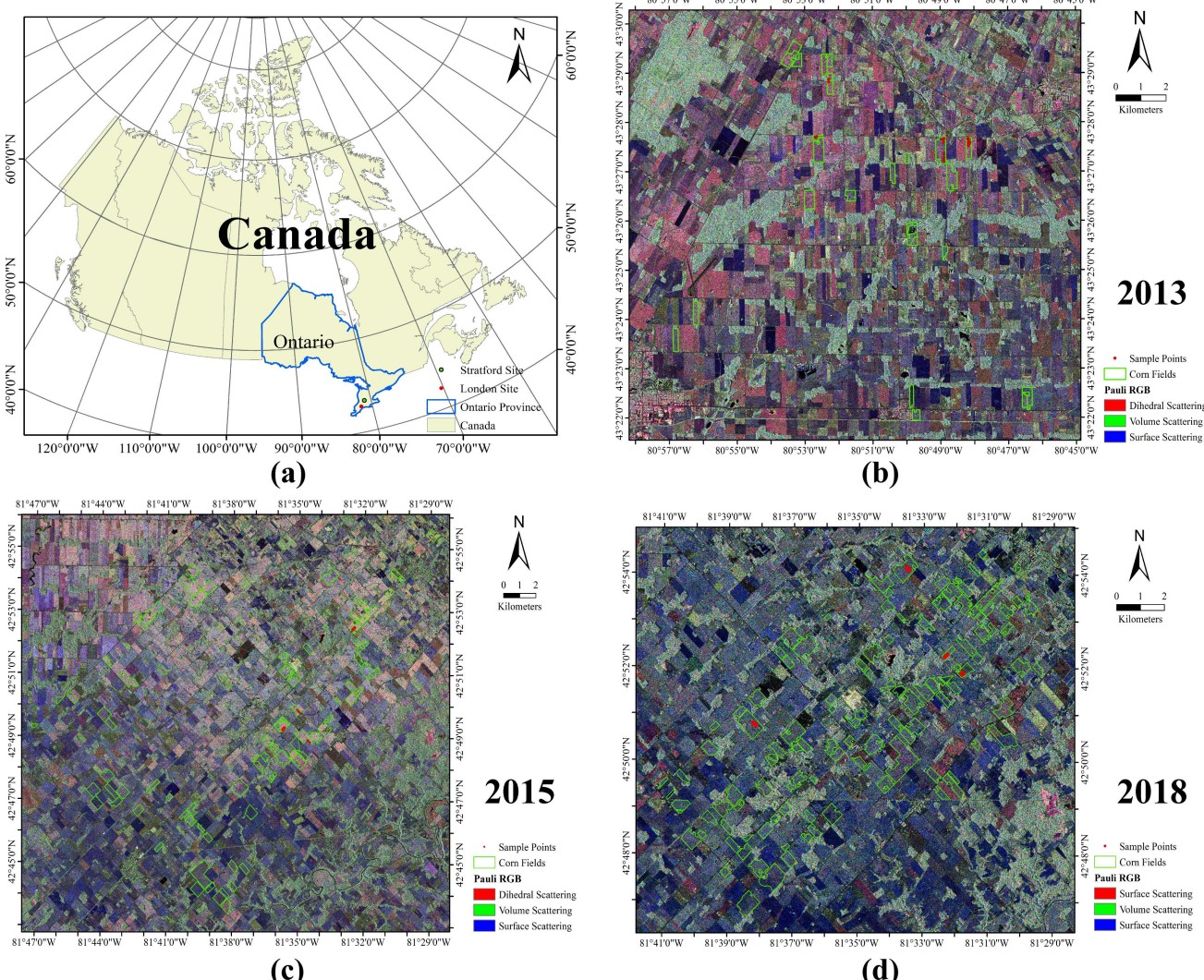

**Figure 1.** Locations and Pauli RGB images of the two study sites (Stratford and London). (**a**) Geographical locations of the two study sites; (**b**) Pauli RGB image acquired on 26 June 2013 at the Stratford site; (**c**) Pauli RGB image acquired on 23 June 2015 at the London site; (**d**) Pauli RGB image acquired on 1 July 2018 at the London site. All boundaries of cornfields and sample points for collecting ground measurements are highlighted in green and red, respectively. (RADARSAT-2 Data and Products © MacDonald, Dettwiler and Associates Ltd. (2013, 2015, 2018)—All Rights Reserved. RADARSAT is an official trademark of the Canadian Space Agency).

In total, 19 scenes of Fine Quad-Pol Wide (FQW) RADARSAT-2 data acquired in 2013, 2015, and 2018 were employed in this study. More specifically, eight scenes were acquired over the Stratford site in 2013 covering the whole corn growing period. Four scenes in 2015 and seven scenes in 2018 were acquired over the London site, which mainly covered the late growth stages. Table 1 shows the acquisition dates, beam modes, incidence angles, resolutions, and orbits of all RADARSAT-2 images.

**Table 1.** RADARSAT-2 Images and Ground Data Acquired for 2013, 2015, and 2018.

| Date | Mode | Incidence | Resolution | Orbit | Fieldwork Date | Number of Corn Sample Points | Average Corn Height (cm) | Study Site |
|---|---|---|---|---|---|---|---|---|
| 23 May 2013 | FQ9W | 27.2 ~ 30.5 | 5.1 × 4.7 | Ascending | 24 May 2013 | 4 | 5.75 | |
| 2 June 2013 | FQ19W | 37.7 ~ 40.4 | 4.7 × 4.7 | Ascending | 4 June 2013 | 16 | 10.06 | |
| 16 June 2013 | FQ9W | 27.2 ~ 30.5 | 5.1 × 4.7 | Ascending | 16 June 2013 | 17 | 25.13 | |
| 26 June 2013 | FQ19W | 37.7 ~ 40.4 | 4.7 × 4.7 | Ascending | 24 June 2013/ 25 June 2013 | 17 | 59.87 | Stratford |
| 10 July 2013 | FQ9W | 27.2 ~ 30.5 | 5.1 × 4.7 | Ascending | 10 July 2013 | 17 | 142.29 | |
| 20 July 2013 | FQ19W | 37.7 ~ 40.4 | 4.7 × 4.7 | Ascending | 21 July 2013 | 11 | 214.35 | |
| 3 August 2013 | FQ9W | 27.2 ~ 30.5 | 5.1 × 4.7 | Ascending | 3 August 2013 | 13 | 254.84 | |
| 13 August 2013 | FQ19W | 37.7 ~ 40.4 | 4.7 × 4.7 | Ascending | 13 August 2013/ 14 August 2013 | 17 | 260.78 | |
| 23 June 2015 | FQ10W | 28.4 ~ 31.6 | 5.5 × 4.7 | Ascending | 23 June 2015 | 25 | 88.44 | |
| 10 August 2015 | FQ10W | 28.4 ~ 31.6 | 5.5 × 4.7 | Ascending | 11 August 2015 | 6 | 266.61 | |
| 3 September 2015 | FQ10W | 28.4 ~ 31.6 | 5.5 × 4.7 | Ascending | 3 September 2015 | 6 | 265.61 | |
| 13 September 2015 | FQ20W | 38.6 ~ 41.3 | 5.1 × 4.7 | Ascending | 13 September 2015 | 6 | 276.72 | |
| 1 July 2018 | FQ10W | 28.4 ~ 31.6 | 5.5 × 4.7 | Ascending | 4 July 2018 | 24 | 182.07 | |
| 25 July 2018 | FQ10W | 28.4 ~ 31.6 | 5.5 × 4.7 | Ascending | 25 July 2018 | 32 | 252.76 | London |
| 1 August 2018 | FQ5W | 22.5 ~ 26.0 | 5.0 × 4.7 | Ascending | 2 August 2018 | 32 | 275.22 | |
| 18 August 2018 | FQ10W | 28.4 ~ 31.6 | 5.5 × 4.7 | Ascending | 18 August 2018 | 32 | 267.77 | |
| 25 August 2018 | FQ5W | 22.5 ~ 26.0 | 5.0 × 4.7 | Ascending | 25 August 2018 | 8 | 214.99 | |
| 1 September 2018 | FQ1W | 17.2 ~ 21.2 | 4.8 × 4.7 | Ascending | 1 September 2018 | 32 | 267.04 | |
| 15 September 2018 | FQ9W | 27.3 ~ 30.5 | 5.1 × 4.7 | Descending | 11 September 2018 | 32 | 267.22 | |

For each year, ground measurements including crop height, crop type, ground photos, soil moisture, and crop phenological stage were recorded. Due to the limitations of weather conditions, human resources, and other reasons, the in situ fieldwork was sometimes not conducted on the exact dates that the RADARSAT-2 satellite overpassed. Since the maximum offset of time was only three days (which occurred on 1 July 2018), it was reasonable to assume that the corn heights did not change from the dates of satellite acquisitions to the ground campaign dates. As it is shown in Table 1, the final number of sample points for each image was different because of limitations due to weather conditions, human resources, and other logistical reasons. For the field campaign in 2013, 17 sample points per day within a maximum of five corn fields were selected for collecting ground measurements. Finally, 112 corn sample points over the Stratford site were collected. For the ground campaign in 2015, 25 sample points per day at a maximum of four cornfields were selected to conduct measurements. In total, 43 corn sample points over the London site were collected. For the ground campaign in 2018, 32 sample points per day in a maximum of four cornfields were exploited. In total, 192 corn sample points over the London site were collected. In summary, a total of 347 corn sample points were collected from the multi-year fieldwork campaigns. Three replicate height readings per sample point were carried out in 2013 and 2015, while twelve readings were conducted in 2018. The measured corn height had a wide range, with values between 3.5 cm and 333.75 cm. The average value of corn height of the ground samples on each fieldwork date ranged from 5.75 cm to 276.72 cm.

*2.2. Polarimetric Observables*

For a fully polarimetric SAR system, the acquired single look complex (SLC) data in H-V polarization basis can be represented by a 2 × 2 scattering matrix, i.e., [61,62],

$$S = \begin{bmatrix} S_{HH} & S_{HV} \\ S_{VH} & S_{VV} \end{bmatrix} \quad (1)$$

where $S_{ij}$ ($i, j = H$ *or* $V$) represents the scattering coefficient from transmitted polarization $i$ and received polarization $j$. The scattering matrix $S$ is used to describe a "pure single target" or deterministic target. For distributed targets, typical in natural media, the second-order statistics (covariance matrix or coherency matrix) are usually exploited to conduct polarimetric analysis. Under the assumption of reciprocal scattering ($S_{HV} = S_{VH}$), the lexicographic basis vector and the Pauli basis vector, respectively, can be expressed as [61,62]

$$k_l = \begin{bmatrix} S_{HH} & \sqrt{2}S_{HV} & S_{VV} \end{bmatrix}^T \tag{2}$$

$$k_P = \frac{1}{\sqrt{2}} \begin{bmatrix} S_{HH} + S_{VV} & S_{HH} - S_{VV} & 2S_{HV} \end{bmatrix}^T \tag{3}$$

Then, the corresponding covariance matrix $C$ and the coherency matrix $T$ are given as [61,62]

$$C = \left\langle k_l k_l^{*T} \right\rangle = \frac{1}{2} \begin{bmatrix} \left\langle |S_{HH}|^2 \right\rangle & \sqrt{2}\left\langle S_{HH}S_{HV}^* \right\rangle & \left\langle S_{HH}S_{VV}^* \right\rangle \\ \sqrt{2}\left\langle S_{HV}S_{HH}^* \right\rangle & 2\left\langle |S_{HV}|^2 \right\rangle & \sqrt{2}\left\langle S_{HV}S_{VV}^* \right\rangle \\ \left\langle S_{VV}S_{HH}^* \right\rangle & \sqrt{2}\left\langle S_{VV}S_{HV}^* \right\rangle & \left\langle |S_{VV}|^2 \right\rangle \end{bmatrix} \tag{4}$$

$$
\begin{aligned}
T &= \left\langle k_P k_P^{*T} \right\rangle \\
&= \frac{1}{2} \begin{bmatrix} \left\langle |S_{HH} + S_{VV}|^2 \right\rangle & \left\langle (S_{HH} + S_{VV})(S_{HH} - S_{VV})^* \right\rangle & 2\left\langle (S_{HH} + S_{VV})S_{HV}^* \right\rangle \\ \left\langle (S_{HH} - S_{VV})(S_{HH} + S_{VV})^* \right\rangle & \left\langle |S_{HH} - S_{VV}|^2 \right\rangle & 2\left\langle (S_{HH} - S_{VV})S_{HV}^* \right\rangle \\ 2\left\langle S_{HV}(S_{HH} + S_{VV})^* \right\rangle & 2\left\langle S_{HV}(S_{HH} - S_{VV})^* \right\rangle & 4\left\langle |S_{HV}|^2 \right\rangle \end{bmatrix}
\end{aligned} \tag{5}
$$

Based on these polarimetric observation matrices, a large number of polarimetric parameters can be extracted for crop monitoring applications [47,48,63,64]. According to their wide usage in crop monitoring studies, a total of 27 polarimetric observables were chosen in this study, as listed in Table 2. First, as the basic products provided by a fully polarimetric SAR system, radar backscattering coefficients in co-polar channels (HH, VV) and cross-polar channel (HV) were chosen, which corresponded to the diagonal elements (C11, C22, C33) in the covariance matrix. Due to their clear physical interpretation in terms of scattering mechanisms, radar backscattering coefficients in the Pauli channels were considered, which could be extracted from the coherency matrix (T11, T22). The widely used total backscattering power, SPAN, was also selected, which was extracted from the sum of the diagonal elements in either the covariance matrix or the coherency matrix (C11 + C22 + C33 or T11 + T22 + T33). The correlation and phase between polarimetric channels (in both the linear and the Pauli basis) were also exploited, which provided additional information about the scattering from the scene. In past studies, these observables have proven very useful for crop phenology monitoring and crop state detection based on multi-temporal analysis of the radar polarimetric response [48,63,65–67]. They provided four complex correlation coefficients, which resulted in eight real observables (amplitudes and phases). Moreover, three backscattering ratios between different linear channels (HH/VV, HV/HH, and HV/VV) were also considered, which had proven to be sensitive to target characteristics. A widely used approach for generating polarimetric features is polarimetric target decomposition, which can be generally categorized as either coherent polarimetric decomposition or incoherent decomposition [61,62]. Due to the capacity to describe distributed targets, incoherent polarimetric decomposition approaches are more suitable for interpreting most nature targets [61,62]. They can be further divided into model-based decomposition and eigenvector-eigenvalue based decomposition. As the pioneering and one of the most popular model-based decomposition methods, the Freeman-Durden three-component decomposition generates three scattering power parameters representing surface, double-bounce, and volume scattering mechanisms, respectively (Freeman and Durden, 1998), which were selected in this study. Additionally, three polarimetric parame-

ters with clear physical meanings from the representative eigenvector-eigenvalue based decomposition proposed by Cloude and Pottier were also used, including the polarimetric scattering entropy H (the degree of scattering randomness in the scattering medium), the alpha angle $\alpha$ (the average dominant scattering mechanism), and the polarimetric anisotropy A (the relative importance between the second and the third scattering mechanism) [68].

**Table 2.** List of 27 polarimetric observables selected in this study.

| Polarimetric Observable | Description |
|---|---|
| C11, C22, C33, | Backscattering coefficients in the linear polarization channels |
| T11, T22 | Backscattering coefficients in the Pauli polarization channels |
| SPAN | Total backscattering power |
| $|\rho_{HHVV}|$, $|\rho_{HVVV}|$, $|\rho_{HHHV}|$, $|\rho_{HH+VV,HH-VV}|$ | Correlation between polarimetric channels |
| $\phi_{HHVV}$, $\phi_{HVVV}$, $\phi_{HHHV}$, $\phi_{HH+VV,HH-VV}$ | Phase difference between polarimetric channels |
| HH/VV, HV/HH, HV/VV | Backscattering ratios |
| $P_s$, $P_d$, $P_v$ | Scattering Power from different scattering mechanisms derived from Freeman-Durden decomposition |
| H, A, $\alpha$ | Entropy, anisotropy, alpha angle from Cloude-Pottier decomposition |
| $|\delta|$, $\phi_\delta$, $\tau$ | Magnitude and phase of the particle scattering anisotropy, the degree of orientation randomness derived from Neumann decomposition |
| RVI | Radar Vegetation Index |

Another model-based decomposition, Neumann decomposition, is aimed at describing vegetation scattering by considering the morphological characteristics of vegetation in scattering modeling [36,69,70]. It has shown the potential advantage of identifying more types of volume scenes [71]. Moreover, two of its output parameters have proven to provide physical meanings similar to Cloude-Pottier decomposition outputs [36,69,70]. A recent study has shown that the third parameter in the Neumann decomposition, the phase of the particle scattering anisotropy, is more effective in improving the classification accuracy with respect to the Cloude-Pottier decomposition [72]. Therefore, the three output parameters from the Neumann decomposition were employed. In addition, the radar vegetation index (RVI) [73] has shown high sensitivity to crop morphological features and thus was also considered in this study.

For each RADARSAT-2 image, a series of preprocessing steps including calibration, speckle filter, and geocoding was conducted. Sigma naught values were obtained by the calibration. A $9 \times 9$ boxcar filter was applied to reduce the speckle noise. Then, a coherency matrix was generated at each pixel. A Digital Elevation Model (PDEM) of the Ontario province, Canada, with a spatial resolution of 30 m was used to geocode the coherency matrix of each image in the Universal Transverse Mercator (UTM) geographic reference. Afterward, 27 polarimetric features listed in Table 2 were extracted for each image. The spatial resolution of all the polarimetric features was the same because they followed the same processing. The final pixel spacing in the geocoded products was 10 m. Furthermore, the corresponding feature vectors, for a total of 347 sample points during the whole RADARSAT-2 acquisition period, were obtained based on the geolocation records of sample points.

### 2.3. Machine Learning Method Used in This Study

Due to the high capacity of prediction, machine learning methods are frequently used for classification and regression problems in remote sensing studies [74,75]. In particular, support vector regression (SVR) and random forest regression (RFR) are two representative examples in this domain and thus were considered for crop height retrieval in this study.

### 2.3.1. Support Vector Regression (SVR)

SVR is the application of a well-known support vector machine model in regression, which has been used in agricultural biophysical parameter estimation with remote sensing data [76–78]. The core idea of SVR is to find an optimal approximating hyperplane to distinguish the input vectors and the predictor variables based on training data, which could be determined by solving a convex optimization problem. Theoretically, it is designed to establish an optimal linear separator and hence is suitable for linear data distribution. However, SVR can also handle nonlinear data distributions after embedded into a kernel framework. With a kernel function, the training data are nonlinearly transformed from the original space to a higher dimensional feature space. In this new space, the new data are better conditioned to make use of a linear separator.

In this study, an IDL-based tool, named imageSVM developed as a non-commercial product at the Geomatics Lab of Humboldt-Universität zu Berlin, specifically designed for support vector machine classification and regression analysis of remote sensing data, was used to carry out the SVR analysis. The radial basis function kernel, called the RBF kernel or the Gaussian kernel, was adopted in the regression model. The kernel parameter, the regularization parameter, and the Epsilon loss function parameter were required to be set to parameterize the SVR. A cross-validation strategy to tune these three parameters was used to reduce model overfitting.

### 2.3.2. Random Forest Regression (RFR)

Unlike SVR, RFR is an ensemble learning method, which uses the subset of the training data to construct a set of decision trees and adopts various non-parametric predictive models [79]. Like the morphological structure of a real tree, a decision tree includes a root node, multiple internal nodes (splits), and various terminal nodes (leaves). Starting from the root to a leaf, a set of decision rules is applied to subdivide the training data into smaller subsets. The predictor variable is assigned as the leaf node. A strategy of bootstrap sampling with replacement is used in RFR to create each individual decision tree. The excluded samples, called out-of-bag samples, are used for model validation. The final prediction is generated by averaging the results from individual decision trees to obtain better prediction performance. In addition, a beneficial property of RFR is that it can also provide estimates of variable importance in the regression, which allows for a better understanding of the sensitivity of the input features to the predictor variable.

In this study, an IDL-based tool, imageRF [80], particularly designed for random forest classification and regression analysis of remote sensing image data, was used to carry out the RFR analysis. The number of decision trees was set to 200 based on the overall consideration of the prediction accuracy and computation time. In the bootstrap sampling for each decision tree, one-third of the training data were set as out-of-bag samples for independent validation. The number of randomly selected features at each split node was determined by the square root of all features.

### 2.3.3. Experimental Design

From Table 1, it is evident that the corn was very short on 23 May, 2 June, and 16 June in 2013. The main backscattering contribution came from the soil, influenced by soil residual and tillage on these dates. The ground photos corresponding to these dates are shown in Figure 2. For this reason, we first carried out the tests by excluding these three observations. Consequently, 16 RADARSAT-2 images and 310 corn sample points were used. In addition, we performed tests involving these three images and all sample points (i.e., 19 RADARSAT-2 images and 347 sample points) and compared the results. In order to construct a regression model, training samples were required for model calibration. For this purpose, the sample points collected from multi-year fieldwork campaigns were randomly divided into two parts. A portion of the samples was dedicated to training samples for model calibration, while the remaining samples were assigned as testing samples for model validation. In our study, 80% of samples were used for calibration and 20% for validation.

It is a common fact that the calibration and prediction accuracies of the regression model can be affected by features of training and testing samples including their distribution and numbers. In order to simulate more random scenarios and investigate the uncertainty of the accuracy, a strategy of bootstrap sampling with replacement was used. As some training samples may have been recycled using this strategy, samples were generated multiple times by random sampling to reduce bias in height estimation [81]. The entire sample points including both short and tall corn heights were considered in the bootstrap sampling method. In total, 10 realizations of random sampling (hereafter named as scenarios) were carried out for generating different datasets for training and testing.

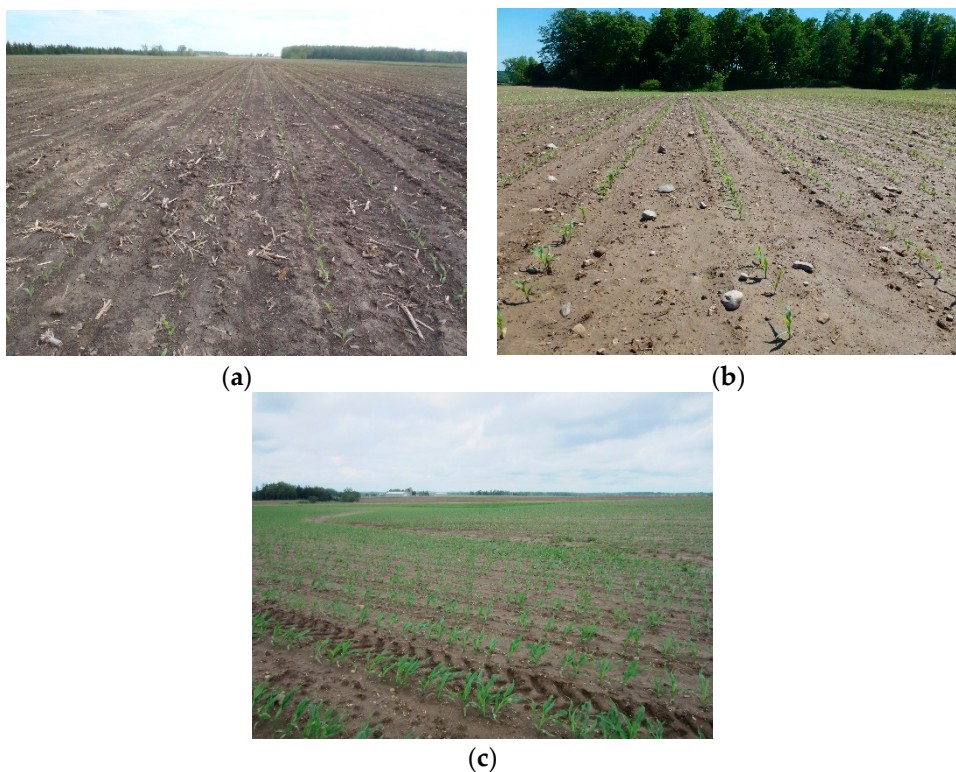

**Figure 2.** Field photos of corn in 2013. (**a**) 25 May; (**b**) 4 June; (**c**) 16 June.

### 3. Results.

*3.1. Comparison between SVR and RFR*

For each dataset, SVR and RFR were conducted. The statistical analysis of model calibration and validation for each dataset with both regression algorithms is shown in Table 3. As expected, results appear varied in different scenarios, which can be attributed to the dependence of the regression model accuracies on the training and testing sets. The results of RFR were generally better than the ones of SVR. RFR obtained overall lower values of root mean square error (RMSE) and higher values of Pearson correlation coefficient (R), despite an opposite behavior for scenario 8. For model calibration, the differences between the performances of both regression algorithms were notable (average RMSE = 22.36 cm for RFR and RMSE = 44.12 cm for SVR), whereas small differences were observed for model validation (average RMSE = 50.40 cm for RFR and RMSE = 54.69 cm for SVR).

**Table 3.** Statistics of model calibration and validation for 10 different datasets with Support Vector Regression (SVR) and Random Forest Regression (RFR) in case of using all 27 polarimetric variables.

| Scenario | Model Calibration | | | | Model Validation | | | |
|---|---|---|---|---|---|---|---|---|
| | SVR | | RFR | | SVR | | RFR | |
| | RMSE (cm) | R | RMSE (cm) | R | RMSE (cm) | R | RMSE (cm) | R |
| 1 | 42.05 | 0.84 | 22.15 | 0.98 | 56.61 | 0.64 | 52.81 | 0.74 |
| 2 | 43.04 | 0.82 | 23.01 | 0.98 | 49.64 | 0.76 | 48.73 | 0.85 |
| 3 | 51.14 | 0.75 | 22.65 | 0.98 | 51.35 | 0.75 | 49.27 | 0.82 |
| 4 | 43.10 | 0.82 | 22.53 | 0.98 | 49.62 | 0.76 | 49.83 | 0.82 |
| 5 | 41.27 | 0.84 | 21.95 | 0.98 | 58.49 | 0.64 | 51.73 | 0.78 |
| 6 | 41.28 | 0.84 | 21.98 | 0.98 | 58.49 | 0.64 | 51.82 | 0.78 |
| 7 | 50.37 | 0.75 | 22.19 | 0.98 | 54.17 | 0.75 | 50.93 | 0.80 |
| 8 | 42.59 | 0.83 | 22.35 | 0.98 | 48.92 | 0.81 | 51.10 | 0.80 |
| 9 | 43.08 | 0.82 | 22.28 | 0.98 | 49.63 | 0.76 | 48.82 | 0.84 |
| 10 | 43.30 | 0.82 | 22.51 | 0.98 | 49.56 | 0.76 | 48.99 | 0.84 |
| Average | 44.12 | 0.81 | 22.36 | 0.98 | 54.69 | 0.73 | 50.40 | 0.81 |

In addition to the statistical indices shown in Table 3, Figure 3 illustrates the scatter plots of measured and predicted corn height obtained with both SVR and RFR methods in scenario 2, in which the regression produced the overall best accuracies. The RFR results exhibited a higher correlation than the results from SVR in both model calibration (see Figure 3a,c) and model validation (see Figure 3b,d). In detail, the model of RFR, in general, yielded overestimated values for lower corn height, while underestimation was observed for higher corn height (taller than around 225 cm). The SVR model generated overestimation and underestimation results for either lower corn height or for higher corn height, while a larger underestimation appeared for higher corn height.

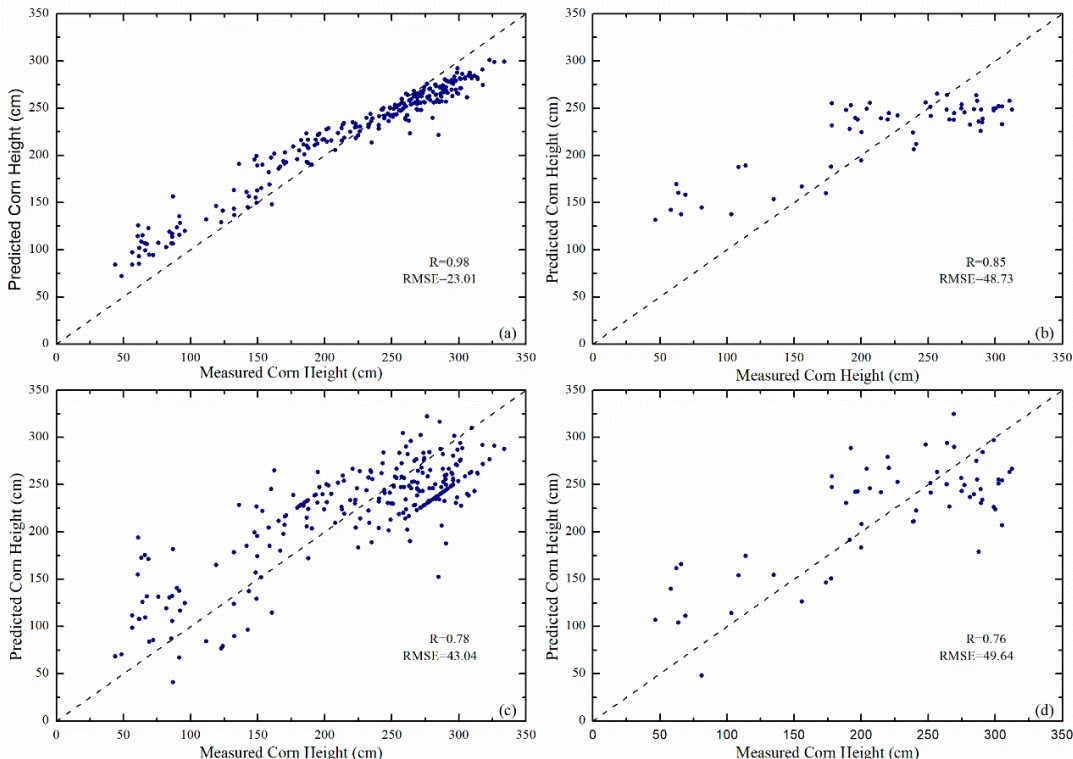

**Figure 3.** Comparison between measured and estimated corn height with scenario 2 in Table 3: (**a**) calibration of the Random Forest Regression (RFR) model; (**b**) validation of the RFR model; (**c**) calibration of the Support Vector Regression (SVR) model; (**d**) validation of the SVR model.

### 3.2. Normalized Variable Importance of RFR

As explained above, the Random Forest algorithm can provide the relative importance of different input variables to the classification or regression accuracy. Therefore, this interesting property of RFR was been also used for further analysis.

Since RFR was carried out under 10 different scenarios and 27 input polarimetric observables were selected in our study, the output values of normalized variable importance of RFR regression for each scenario were variable. It is difficult and unfair to analyze the variable importance using values from one specific scenario. For this reason, Figure 4 shows the ranking of the importance of each polarimetric observables in the regression based on the average values of output normalized variable importance for the 10 scenarios. For better visualization, the parameters belonging to the same or similar group are marked with the same color. The contribution of the double-bounce component (Pd) from the Freeman-Durden decomposition was the most important variable among the selected 27 polarimetric observables. It demonstrated a strong correlation between double-bounce scattering and crop height. Moreover, considering the polarimetric decompositions used in this study, parameters from the Freeman-Durden decomposition appeared to be more relevant than parameters from the Neumann and Cloude-Pottier decompositions. The reasons may be attributed to the nature of the Freeman-Durden decomposition models, which are physically based rather than purely mathematical as in the Cloude-Pottier decomposition. Hence, its applicability is more general than the Neumann decomposition that assumes the dominance of volume scattering, which is not always the case for crops. In particular, the contribution of the volume scattering component (Pv) from the Freeman-Durden decomposition took second place in the variable importance ranking, just after Pd. Notably, the polarimetric anisotropy contributed more entropy and alpha angle from Cloude-Pottier decomposition.

The magnitude of the degree of orientation randomness of the particle scattering anisotropy was the most important variable from the Neumann decomposition. Among the backscattering coefficients, C22 played the most important role, which was highly related to the volume scattering component Pv, and took third place in the ranking. C33, which represented the VV polarization, showed less contribution and was placed in the last position among the linear backscattering coefficients. T22 was the second most important, and was highly related to the double-bounce scattering component Pd, which took fourth place. The total scattering power SPAN ranked behind T22 but was still at the front position. Among the backscattering ratios, HV/HH was the most important parameter, and the corresponding contribution was significantly larger than the other two linear ratios. Although co-polar correlation magnitude $|\rho_{HHVV}|$ took up a middle position similar to RVI in the ranking, most correlation magnitudes and polarimetric phases contributed less to the regression and were found at the end of the importance list. From the aforementioned analysis, it was clear to see that SAR parameters related to the double bounce and volume scattering components (e.g., Pd, Pv, C22, T22) showed high sensitivity to crop height and strongly drove height estimation for the RFR method. The reasons may be attributed to the nature of the scattering mechanisms which interact differently with different plant structures (e.g., stem, flower, leaf, tassel) as the crop development advances. For example, double-bounce and volume scattering components vary significantly with crop growth stages, which generally are low in the early stage and high in later development stages (e.g., stem elongation, tassel, and stigmata emergence).

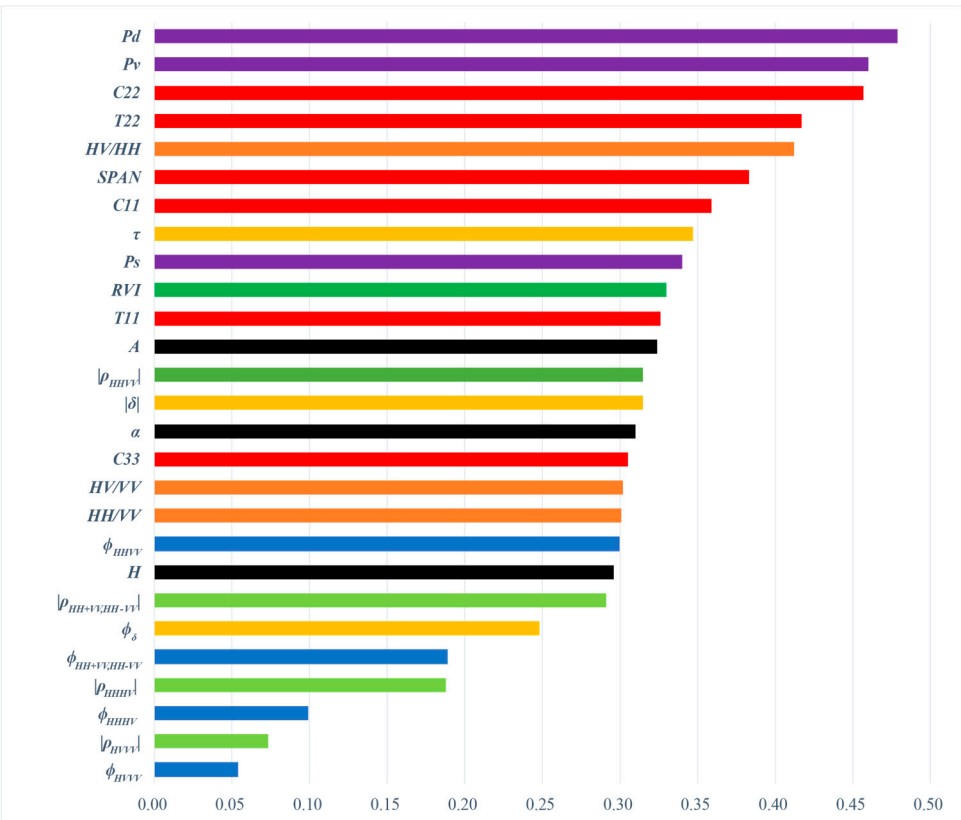

**Figure 4.** Normalized variable importance of RFR in regression for corn height based on averaged output values of 10 scenarios. Different types of polarimetric variables are represented as follows: Backscattering coefficients and SPAN (red), Correlation magnitude (light green), Polarimetric phase (blue), Backscattering ratios (orange), Freeman-Durden decomposition (purple), Cloude-Pottier decomposition (black), Neumann Decomposition (yellow) and RVI (green). A description of how variables were named can be found in Table 2.

## 4. Discussion

### 4.1. Tests with Fewer Polarimetric Observables

It is well known that using more features does not always generate better regression accuracies because of redundant or correlated information. Therefore, it is interesting to further check the regression accuracies in case fewer polarimetric observables can be used. From the diagram of variable importance ranking shown in Figure 3, the contributions of the first six polarimetric variables are obviously larger than the rest of the variables. Therefore, additional RFR tests were conducted by using only the first six polarimetric variables (i.e., Pd, Pv, C22, T22, HV/HH, and SPAN). At the same time, corresponding SVR tests were also carried out for comparison purposes. The statistical analyses of model calibration and validation for 10 different scenarios with SVR and RFR are presented in Table 4. It is apparent that regression estimation accuracies of SVR and RFR improve with respect to the previous results for each scenario in Table 3, even though sometimes model calibration accuracies decrease slightly. As in the previous results, RFR provides better accuracies than SVR in most scenarios. The differences between the two machine learning regression methods are smaller in the case of using fewer polarimetric variables. The average values of RMSE and R with RFR prediction are 47.76 cm and 0.79, while the corresponding values are 47.90 cm and 0.78 with SVR prediction, respectively. The best regression estimation results were produced in scenario 1 with RFR. The corresponding values of RMSE and R reach 42.69 cm and 0.84, respectively. To further analyze the results in scenario 1, the scatterplots of measured and estimated corn height with SVR and RFR methods are presented in Figure 5. The patterns of model calibration are similar to the ones shown in Figure 3. The overall distributions of RFR and SVR estimation results show

similar patterns, in which overestimation is mostly observed for lower corn height, and underestimation is observed for taller corn height. To analyze the SVR and RFR results at different stages of corn growth and maintain sufficient samples for statistics, the simple division method for growth stages of corn proposed in [48] was adopted. The corn height of less than 150 cm is addressed as the early stage, and height that is taller than 150 cm is defined as the later stage [48]. After calculation of values of RMSE and R, statistics of model validation at an early stage and later stage for 10 different scenarios with SVR and RFR are presented in Table 5. It is clear to see that the results of SVR and RFR are better at the later stage than the ones at the early stage.

**Table 4.** Statistics of model calibration and validation for 10 different datasets with SVR and RFR in case of using the top six polarimetric variables in the variable importance ranking.

| Scenario | Model Calibration | | | | Model Validation | | | |
| --- | --- | --- | --- | --- | --- | --- | --- | --- |
| | SVR | | RFR | | SVR | | RFR | |
| | RMSE (cm) | R | RMSE (cm) | R | RMSE (cm) | R | RMSE (cm) | R |
| 1 | 46.42 | 0.79 | 22.33 | 0.97 | 45.45 | 0.79 | 42.69 | 0.84 |
| 2 | 49.14 | 0.76 | 22.69 | 0.97 | 48.04 | 0.78 | 47.81 | 0.79 |
| 3 | 45.84 | 0.79 | 21.91 | 0.97 | 50.75 | 0.74 | 48.07 | 0.78 |
| 4 | 45.73 | 0.79 | 22.04 | 0.97 | 46.20 | 0.80 | 48.12 | 0.79 |
| 5 | 45.34 | 0.81 | 21.53 | 0.97 | 53.89 | 0.72 | 50.68 | 0.74 |
| 6 | 48.69 | 0.77 | 21.28 | 0.97 | 51.57 | 0.73 | 51.29 | 0.73 |
| 7 | 45.81 | 0.80 | 22.42 | 0.97 | 45.38 | 0.81 | 46.14 | 0.82 |
| 8 | 45.82 | 0.80 | 22.44 | 0.97 | 45.38 | 0.81 | 47.07 | 0.81 |
| 9 | 45.71 | 0.79 | 22.31 | 0.97 | 46.14 | 0.80 | 47.72 | 0.79 |
| 10 | 45.74 | 0.79 | 22.36 | 0.97 | 46.22 | 0.80 | 48.01 | 0.79 |
| Average | 46.42 | 0.79 | 22.13 | 0.97 | 47.90 | 0.78 | 47.76 | 0.79 |

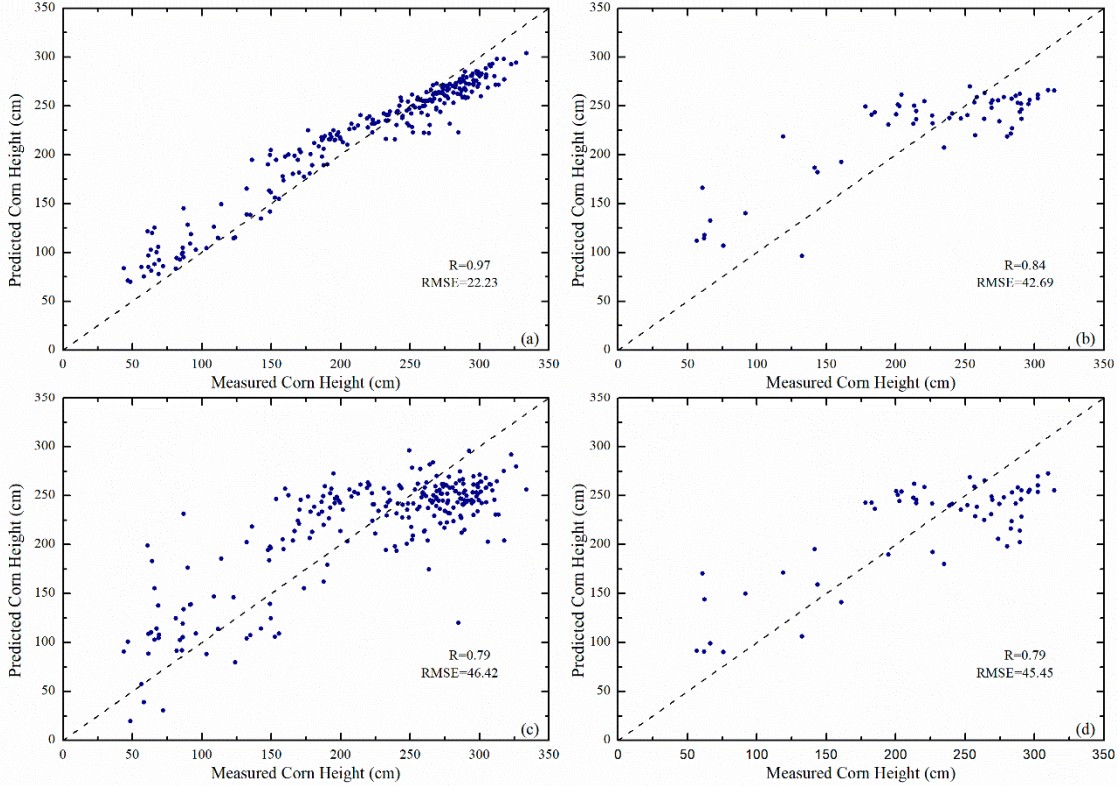

**Figure 5.** Comparison between measured and estimated corn height with scenario 1 in Table 4: (**a**) calibration of the RFR model; (**b**) validation of the RFR model; (**c**) calibration of the SVR model; (**d**) validation of the SVR model.

**Table 5.** Statistics of model validation at two different growing stages for 10 different datasets with SVR and RFR in case of using the top six polarimetric variables in the variable importance ranking.

| Scenario | Model Validation | | | | | | | |
| | SVR (Height < 150 cm) | | SVR (Height > 150 cm) | | RFR (Height < 150 cm) | | RFR (Height > 150 cm) | |
| | RMSE (cm) | R | RMSE (cm) | R | RMSE (cm) | R | RMSE (cm) | R |
|---|---|---|---|---|---|---|---|---|
| 1 | 53.91 | 0.52 | 43.41 | 0.27 | 62.02 | 0.51 | 37.23 | 0.40 |
| 2 | 59.61 | 0.16 | 45.16 | 0.32 | 49.19 | 0.64 | 47.51 | 0.26 |
| 3 | 44.46 | 0.80 | 52.01 | 0.10 | 48.47 | 0.92 | 47.98 | 0.10 |
| 4 | 51.75 | 0.34 | 44.91 | 0.35 | 50.71 | 0.58 | 47.55 | 0.26 |
| 5 | 55.13 | 0.76 | 53.49 | 0.31 | 68.37 | 0.72 | 43.55 | 0.30 |
| 6 | 60.29 | 0.67 | 48.46 | 0.35 | 66.06 | 0.71 | 45.58 | 0.26 |
| 7 | 59.99 | 0.37 | 41.99 | 0.47 | 64.46 | 0.48 | 41.71 | 0.41 |
| 8 | 59.99 | 0.37 | 41.99 | 0.47 | 64.59 | 0.49 | 42.89 | 0.38 |
| 9 | 51.76 | 0.34 | 44.84 | 0.35 | 49.00 | 0.60 | 47.44 | 0.28 |
| 10 | 51.75 | 0.34 | 44.94 | 0.35 | 51.06 | 0.63 | 47.33 | 0.28 |
| Average | 54.86 | 0.47 | 46.12 | 0.33 | 57.39 | 0.63 | 44.88 | 0.29 |

Based on the regression model, a corn height map can be generated on each date. For example, the map of estimated corn height on 15 September 2018 is presented in Figure 6. Locations of four cornfields and eight sample points in each field are marked on the map. The estimated corn height ranges between 80 and 295 cm. Considering most cornfields are at the late crop growing stage on that date, the results of estimated height are underestimated which is consistent with the previous analysis. In addition, the values of estimated height somehow show variation among different fields, which can be attributed to the diversity of cornfield conditions, such as soil moisture and roughness, topography, precipitation, and fertilization. Table 6 shows the measured and estimated corn heights in 32 sample points. The difference is very small at some points and large at other points. The calculated RMSE and R values are 32.61 cm and 0.59, respectively.

**Table 6.** The measured and estimated corn heights of 32 sample points on 15 September 2018 based on RFR model prediction.

| Field Name | Corn Height (cm) | 1 | 2 | 3 | 4 | 5 | 6 | 7 | 8 |
|---|---|---|---|---|---|---|---|---|---|
| C1 | Measured | 274.83 | 251.5 | 295.17 | 286.83 | 290.67 | 285.5 | 235.75 | 271.41 |
| | Estimated | 252.55 | 228.26 | 283.05 | 275.69 | 279.13 | 274.72 | 244.21 | 263.61 |
| C2 | Measured | 273.92 | 301 | 371.58 | 260.42 | 241.08 | 283.17 | 298.75 | 283.92 |
| | Estimated | 255.66 | 259.90 | 255.99 | 255.32 | 244.74 | 273.18 | 282.48 | 227.11 |
| C3 | Measured | 255.75 | 251.67 | 219.83 | 248.08 | 162.5 | 220.67 | 253.5 | 204.33 |
| | Estimated | 258.35 | 261.41 | 228 | 232.20 | 199.91 | 254.53 | 248.51 | 261.60 |
| C4 | Measured | 276.25 | 288.50 | 290.75 | 288.08 | 296.58 | 298.92 | 306 | 284.25 |
| | Estimated | 270.46 | 280.46 | 246.22 | 252.87 | 255.87 | 284.98 | 268.49 | 262.39 |
| RMSE (cm) | | | | | 32.61 | | | | |
| R | | | | | 0.59 | | | | |

## 4.2. Tests with All Images Including Very Short Corn Height

In previous tests, the first three images in 2013 are excluded due to their very short corn height. It is interesting to check the performance when all images, including very short corn height, are used. Ten realizations of SVR and RFR tests were carried out again, respectively. As in previous tests, 80% of samples were used for calibration and 20% for validation. However, 19 RADARSAT-2 images and 347 corn samples were used this time. The statistical analysis of model calibration and validation for each dataset with both regression algorithms is shown in Table 7. As expected, the results still show variation in different scenarios and the results of RFR are better than SVR. However, the values of RMSE and R are worse than the results in Table 3 (using 16 RADARSAT-2 images with

310 corn samples). The average values of RMSE and R for RFR in model validation are 54.55 cm and 0.83, and the values for SVR are 56.75 cm and 0.80, respectively. It somehow indicates the limitation of this kind of regression methods, i.e., very short height will affect the accuracy of estimation.

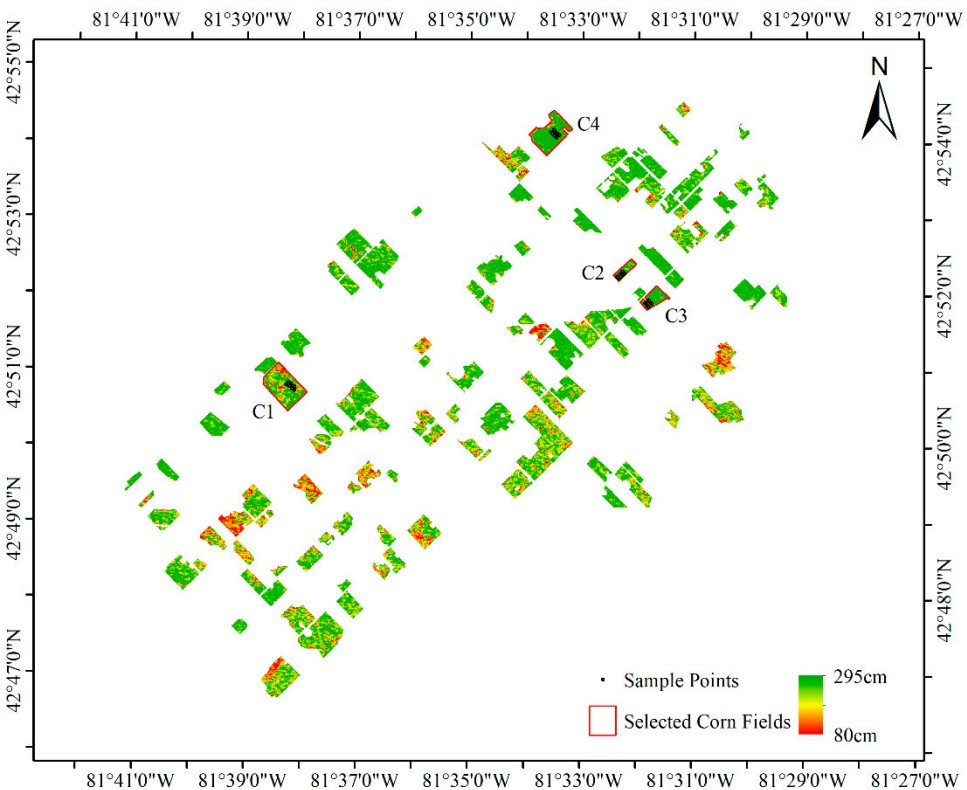

**Figure 6.** Map of corn height on 15 September 2018 based on RFR model prediction.

**Table 7.** Statistics of model calibration and validation for 10 different datasets with SVR and RFR in case of using all 27 polarimetric variables and all 19 RADARSAT-2 images.

| Scenario | Model Calibration | | | | Model Validation | | | |
|---|---|---|---|---|---|---|---|---|
| | SVR | | RFR | | SVR | | RFR | |
| | RMSE (cm) | R | RMSE (cm) | R | RMSE (cm) | R | RMSE (cm) | R |
| 1 | 48.63 | 0.87 | 22.38 | 0.99 | 58.81 | 0.80 | 54.41 | 0.84 |
| 2 | 48.53 | 0.86 | 22.63 | 0.99 | 58.75 | 0.82 | 55.62 | 0.86 |
| 3 | 50.05 | 0.86 | 22.22 | 0.99 | 59.87 | 0.75 | 54.56 | 0.83 |
| 4 | 50.45 | 0.87 | 22.78 | 0.99 | 54.37 | 0.76 | 52.98 | 0.78 |
| 5 | 49.09 | 0.87 | 22.20 | 0.99 | 55.16 | 0.79 | 54.73 | 0.84 |
| 6 | 48.27 | 0.87 | 22.18 | 0.99 | 57.64 | 0.83 | 57.62 | 0.84 |
| 7 | 49.13 | 0.86 | 23.05 | 0.99 | 53.44 | 0.85 | 53.15 | 0.88 |
| 8 | 49.95 | 0.86 | 22.43 | 0.99 | 54.86 | 0.76 | 50.65 | 0.82 |
| 9 | 48.51 | 0.87 | 22.14 | 0.99 | 56.53 | 0.81 | 55.59 | 0.82 |
| 10 | 49.23 | 0.86 | 22.47 | 0.99 | 58.05 | 0.78 | 56.20 | 0.83 |
| Average | 49.18 | 0.87 | 22.45 | 0.99 | 56.75 | 0.80 | 54.55 | 0.83 |

*4.3. Limitations and Future Research*

In this study, two common machine learning techniques used for scientific purposes, i.e., RFR and SVR, were evaluated for crop height estimation of corn from multi-year RADARSAT-2 polarimetric observables. There are some limitations to these methods. Firstly, the methods depend on the availability of a large number of sample data and a

good sample distribution. In our case, although the number of samples is large, corn samples have more values of heights taller than 2 m and fewer values between 1 m and 2 m. This might affect the accuracy of model calibration and estimation. Secondly, the output results somehow show overestimation and underestimation. The results show worse performance at the early growth stages, especially in the case of estimating very short crop height. Applying piecewise regression methods at different growing stages may improve the results. Third, the PolSAR data are acquired in different modes with different incidence angles, which might also influence the accuracy of estimation. Additionally, field conditions such as soil moisture, irrigation, and fertilization, may also affect the estimation results.

Future work will focus on testing these techniques for height estimation of other crop types, such as wheat, soybean, and rice. Moreover, it is worth investigating how changes in crop physiology associated with crop development and precipitation (or irrigation regimes) could contribute to changes in the priority of different PolSAR parameters as the crop matures along the full season. Tests and analysis with PolSAR data at other radar frequencies (such as TerraSAR-X at X band, ALOS-2 at L band) over different sites will also be investigated in future research.

## 5. Conclusions

This study presents a demonstration of crop height retrieval based on space-borne PolSAR data with machine learning techniques. The techniques have been tested with RADARSAT-2 data in cornfields covering the whole corn growing period. The potential of two popular machine learning regression algorithms (SVR and RFR) was investigated, including an identification of the relevant features by means of the normalized variable importance of RFR. A set of 27 representative PolSAR observables was initially selected and analyzed in this work. The results show a satisfactory prediction performance for corn height mapping at a large scale, with RMSE around 40–50 cm considering the whole growth cycle, with corn height over 3 m at late stages. The RFR approach overall outperforms the SVR method in all tests. The best result is generated by the RFR method when selecting a subset of six polarimetric features, with an RMSE of 42.8 cm, which indicates that fewer selected polarimetric features can generate better results than using all features. Regarding the analysis of the relative importance of all polarimetric features considered, results highlight that parameters related to double-bounce and volume scattering are the most important polarimetric features for corn height estimation. In addition, the HV/HH ratio appears to be the most representative among all three backscattering ratios. Compared with past studies on crop height retrieval with SAR data, this research provides a potentially efficient method and a new perspective on the use of PolSAR data.

**Author Contributions:** Conceptualization, Q.X. and J.W.; methodology, Q.X.; software, Q.X.; validation, Q.X. and C.L.; formal analysis, Q.X., J.W. and X.P.; investigation, Q.X. and J.W.; resources, J.W.; data curation, Q.X., C.L., J.W. and J.S.; writing—original draft preparation, Q.X.; writing—review and editing, J.W., J.M.L.-S., J.Z., J.S., X.P. and J.D.B.-B.; visualization, Q.X.; supervision, J.W.; project administration, Q.X. and J.W.; funding acquisition, Q.X., J.W., J.Z., H.F. and J.M.L.-S. All authors have read and agreed to the published version of the manuscript.

**Funding:** This research was funded in part by the National Natural Science Foundation of China (Grant No. 41804004, 41820104005, 41531068, 41904004), the Canadian Space Agency SOAR-E program (Grant No. SOAR-E-5489), the Fundamental Research Funds for the Central Universities, China University of Geosciences (Wuhan) (Grant No. CUG190633), and the Spanish Ministry of Science, Innovation and Universities, State Research Agency (AEI) and the European Regional Development Fund under project TEC2017-85244-C2-1-P.

**Institutional Review Board Statement:** Not applicable.

**Informed Consent Statement:** Not applicable.

**Data Availability Statement:** No new data were created or analyzed in this study. Data sharing is not applicable to this article.

**Acknowledgments:** RADARSAT-2 Data and Products © MacDonald, Dettwiler and Associates Ltd. (2013, 2015, 2018)—All Rights Reserved. RADARSAT is an official trademark of the Canadian Space Agency. The authors would like to thank the Canadian Space Agency and Agriculture and Agri-Food Canada for providing the RADARSAT-2 data, and Yang Song, Yu Bo, Xiaodong Huang from GITA lab, UWO for their help with field work. In addition, the authors acknowledge A&L Canada Inc. for the access to the corn fields.

**Conflicts of Interest:** The authors declare no conflict of interest.

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
