# Peer review of "Crop Height Estimation of Corn from Multi-Year RADARSAT-2 Polarimetric Observables Using Machine Learning"

_remotesensing, doi:10.3390/rs13030392_

Round 1
Reviewer 1 Report
General comments
I thank the authors for putting together a substantial amount of work representing both field campaigns and associated RS acquisitions across several years and crops. This work is in some regards similar to prior studies (e.g., Nasirzadehdizaji et al. 2019 ApplSci) but significantly more statistically focused (not unlike Liao et al. 2017 IJRS) and consequently insightful about the potential for ML methods to reveal the components of SAR acquisitions that are most tied to crop height, and how to use them in a ML modeling framework. The study demonstrates, for a prescribed set of agricultural fields, the accuracy of two ML approaches for predicting crop height. Further, they show for one approach, RFR, the priority among polarimetric parameters in predicting crop height. Some interesting results are included that are relevant to academia, governments and monitoring, and R&D in the ag industry. In particular, I was intrigued by RFR's under and overestimation of crop height as a function of crop height.
Overall the paper is worthy of publication with only minor grammatical changes. However, I will still point out that the paper currently misses an opportunity to discuss, with reasonable depth, considerations of how the developmental changes in crop physiology could contribute to changes in the priority of different SAR parameters as the crop matures. That is, height detection sensitivity no doubt changes with respect to height, which also reflects underlying changes in the structure of the plant (stem, flower, leaf, tassel). This could be particularly true in later stage development (e.g., stem elongation, tassel, and stigmata emergence) and perhaps behind the over and underestimation in the RFR approach. With all that said, I think the paper could be improved by simply elaborating on how the RS physics of double bounce, volume scattering, etc. could be interacting differently with different plant structures as development progresses, and what this means for detection accuracy at different stages. This is especially important given that this study is clearly indicating that some of the double bounce and volume scattering components are strongly driving height prediction for the RFR method.
Specific comments
lines 315-316, I’m not sure if the bootstrap sampling with replacement is discussed as a potential limitation to the results, in that some training samples may have been recycled.
Figure 2, Given that SVR depends so strongly on the given subset of training data and the impact of cost functions and thresholds for points that lie near/beyond the margin, it seems unsurprising that the results in Figure 2(c) seem almost bounded within linear channels. And as stated in lines 337-339, SVR generates results that either over or underestimate corn height, regardless of corn height. i.e., if very tall or short corn is not included in the training subset. Is the sampling subset limitation of SVR discussed elsewhere?
Figure 3, is a significant figure that imparts a great deal of information. The authors did an excellent job here.
Line 409, remove the extra semicolon.
Reviewer 2 Report
The article describes an interesting example of using PolSAR data to solve an agricultural problem of corn height estimation. The machine learning techniques described in this article are common in scientific research. However, the Authors have found very important applications for them. Below are the main comments with line numbers.
29: Is RMSE 40-50 cm is true for all stages of corn growth? In the initial stages, it is a pretty big error.
37: "Crop height is an important agronomic descriptor related to ... crop type ... potential yield ...". Please describe in more detail how accuracy of 40-50 cm can affect the accuracy of yield predictions for a field or a group of fields? And what specific crop types can be distinguished with such precision?
112: "a empirical " - replace with "an empirical"
121-122: "However, to date there are few studies reporting crop height retrieval based on regression with PolSAR observables [52]." - it is desirable to specify more specifically what the shortcoming/problem of these studies is and how you solve this problem. What is the uniqueness of your research (the main difference from the results of other authors).
In general, the Introduction is very extensive and represents a comprehensive analytical review. The material presented in the Introduction allows to dive into the research topic in detail. The Introduction provides comprehensive information about existing methods and relatively little justification for the choice of approach that the Authors used in the article.
175: “crop type” - do you mean different corn types or e.g. corn/soybeans/wheat?
175-176: are the ground measurement results publicly available?
193: Table 1. It would be interesting to know the average corn height in each date.
208: “a total of 27 polarimetric observables” - have you investigated the relationship of each of these variables to corn height? What is the spatial resolution of each variable?
253: Sigma naught values ​​are sensitive to precipitation. Was the influence of precipitation (or irrigation regimes) on the accuracy of calculations taken into account?
383: Figure 3. Is it possible to substantiate the obtained results?
388: 4. Discussion
Please describe the SVR and RFR results at different stages of corn growth. What will be the R and RMSE values at the initial stages of corn growth? In the Discussion section, also please assess the applicability of the described approach to other satellite data. Also please add a paragraph about the possibilities for further development of the described methods and possible approaches to reducing the RMSE. The Discussion section can be supplemented with a table containing Sample Points with actual and predicted corn heights.
400: missing space
409: punctuation error
423: please place on the map or next to Figure 5 the actual and predicted corn heights in Sample Points
